# Parasitism and Suitability of *Trichogramma chilonis* on Large Eggs of Two Factitious Hosts: *Samia cynthia ricini* and *Antheraea pernyi*

**DOI:** 10.3390/insects15010002

**Published:** 2023-12-20

**Authors:** Yue-Hua Zhang, Ji-Zhi Xue, Talha Tariq, Tian-Hao Li, He-Ying Qian, Wen-Hui Cui, Hao Tian, Lucie S. Monticelli, Nicolas Desneux, Lian-Sheng Zang

**Affiliations:** 1Jiangsu Key Laboratory of Sericultural Biology and Biotechnology, School of Biotechnology, Jiangsu University of Science and Technology, Zhenjiang 212100, China; 2National Key Laboratory of Green Pesticide, Key Laboratory of Green Pesticide and Agricultural Bioengineering, Ministry of Education, Guizhou University, Guiyang 550025, China; 3INRAE, UMR ISA, Université Côte d’Azur, 06000 Nice, France; lucie.monticelli@gmail.com (L.S.M.); nicolas.desneux@inrae.fr (N.D.)

**Keywords:** parasitoid, mass rearing, fitness, eri silkworm, biological control

## Abstract

**Simple Summary:**

In China, the large eggs of the Chinese oak silkworm (COS), *Antheraea pernyi,* have been extensively utilized as beneficial factitious hosts for the mass rearing of parasitoid wasps and in the biological control of agricultural and forestry pests. Furthermore, the eri silkworm (ES), *Samia cynthia ricini,* has a widely distributed large egg that has the potential for use in the mass rearing of *Trichogramma* parasitoids. This study investigates the role of *T. chilonis* as a dominant *Trichogramma* species in lepidopteran pest control for agricultural and forestry production. It offers a comparative analysis of *T. chilonis* parasitism and suitability with respect to the large eggs from two factitious hosts, the ES and COS. The results reveal the feasibility of mass producing *T. chilonis* using ES eggs and provide insights into this species’ parasitism of both COS and ES eggs. These findings are crucial for developing a cost-effective strategy for large-scale *Trichogramma* rearing using ES eggs, contributing to efficient biological pest control in agricultural and forestry contexts.

**Abstract:**

*Trichogramma*, an effective biological control agent, demonstrates promise in environmentally sustainable pest management through its parasitic action toward insect eggs. This study evaluates the parasitism fitness and ability of *T. chilonis* with regard to two factitious host eggs, aiming to develop a cost-effective biological control program. While *T. chilonis* demonstrated the ability to parasitize both host eggs, the results indicate a preference for ES eggs over COS eggs. The parasitism and emergence rates of *T. chilonis* regarding ES eggs (parasitism: 89.3%; emergence: 82.6%) surpassed those for COS eggs (parasitism: 74.7%; emergence: 68.8%), with a notable increase in the number of emergence holes observed in the ES eggs compared to the COS eggs. Moreover, the developmental time of *T. chilonis* for ES eggs (10.8 days) was shorter than that for COS eggs (12.5 days), resulting in a lower number of dead wasps produced. Notably, no significant difference was observed in the female ratios between the two species. A comprehensive analysis was conducted, comparing the size and shell thickness of the two factitious hosts. The ES eggs exhibited smaller dimensions (length: 1721.5 μm; width: 1178.9 μm) in comparison to the COS eggs (length: 2908.8 μm; width: 2574.4 μm), with the ES eggshells being thinner (33.8 μm) compared to the COS eggshells (47.3 μm). The different host species had an effect on the body length of the reared parasitoids, with *T. chilonis* reared on COS hosts exhibiting a larger body length (female: 626.9 µm; male: 556.7 µm) than those reared on ES hosts (female: 578.8 µm; male: 438.4 µm). Conclusively, the results indicate that ES eggs present a viable alternative to COS eggs for the mass production of *Trichogramma* species in biological control programs.

## 1. Introduction

Biological control offers an environmentally sustainable alternative for pest management [1]. Biological control methods present an environmentally sustainable alternative for pest control management, distinguished by their ecological compatibility when contrasted with chemical control. When using these approaches, one often encounters challenges such as the emergence of multiple insecticide resistances in pests and the presence of insecticide residues impacting non-target species [2,3]. *Trichogramma* (Hymenoptera: Trichogrammatidae), one of the most important egg parasitoids and natural enemies in biological pest control worldwide, can parasitize the eggs of more than 400 insect species in eight orders and especially prefers to parasitize the eggs of lepidopteran pests [4]. The inundative release of *Trichogramma* can be used to effectively control the striped rice stem borer *Chilo suppressalis* (Walker), the Asian corn borer *Ostrinia furnacalis* (Guenee), and other pests [5,6,7]. Among them, *T. chilonis* has been widely used in the control of major pests such as the Asian corn borer *Ostrinia furnacalis*, the rice leaf roller *Cnaphalocrocis medinalis*, the oriental armyworm *Mythimna separata*, the soybean pod borer *Leguminivora glycinivorella,* and other major pests, and it has provided promising economic and environmental benefits [1]. For example, a study reported that field releases of *T. chilonis* used to control sugarcane stem borer yielded egg parasitism rates ranging from 65% to 94% [8]. In addition, previous reports indicated up to 86% parasitism of *T. chilonis* toward *Spodoptera frugiperda* eggs [9] and revealed that *Trichogramma* parasitoid effectiveness is modulated by egg scale thickness when parasitizing this host [10,11].

The mass production and application of *Trichogramma* are closely linked to the use of factitious host eggs. Initially, the wasps must prioritize parasitizing the factitious host egg. After the parasitoid species lays its eggs, the offspring must consume enough nutrients from the host egg for full growth and development and successful emergence. Currently, the factitious hosts that can be used for the mass rearing of *Trichogramma* include the eggs of the rice moth *Corcyra cephalonica*, the eggs of the Chinese oak silkworm (COS) *Antheraea pernyi*, the eggs of the angoumois grain moth *Sitotroga cerealella* (Olivier), and the eggs of the Mediterranean flour moth *Ephestia kuehniella*. Compared to the eggs of *C. cephalonica*, *S. cerealella*, *E. kuehniella,* and other hosts, COS eggs are inexpensive, have excellent quality, and offer prolonged storage potential. A single egg that produces 1–2 adult *Trichogramma* wasps is called a small egg. An adult egg that produces three or more *Trichogramma* is called a large egg [12]. Large eggs have the advantages of being inexpensive and easily transportable compared to small eggs, so research on rearing *Trichogramma* with large eggs is more significant [1]. This factor makes COS eggs an excellent factitious host for the mass production of *Trichogramma* in China. Additionally, *T. chilonis* can engage in multiparasitism with other *Trichogramma* species to parasitize the eggs of the COS [13,14]. Although *Trichogramma* parasitoids are usually generalists, many species could not successfully reared their offspring using COS eggs, such as *Trichogramma ostrinia*, which can complete its development inside COS eggs but is unable to form an emergence hole due to the thick shells of COS eggs, resulting in the death of the parasitized wasps inside the eggs [15]. Several studies have reported varying suitability of *Trichogramma* for different hosts; characteristics such as host egg size, eggshell thickness, nutrients of the host egg fluid, and host age affect *Trichogramma* parasitism [16,17,18]. These factors often play a crucial role in determining the suitability of factitious hosts for mass production; as a result, the selection of suitable factitious hosts for biological control is of vital importance. In recent years, *T. chilonis* has been mainly reared on small host eggs, for which mass production costs are high and production efficiency is low. However, when rearing parasitoids using COS eggs, problems often occur with regard to unstable parasitism rates and emergence rates, which seriously affect the product quality of *T. chilonis* and the effectiveness of pest control in the field and, at the same time, lead to the wasting of resources in mass production [1]. Since the COS is a unique resource in East Asian countries, including China, North Korea, South Korea, and Japan, its availability limits the widespread adoption of the COS as a factitious host for rearing *Trichogramma* species for agricultural and forestry pest control worldwide [1,19]. 

The Eri silkworm (ES), *Samia cynthia ricini* Drury (Lepidoptera: Saturniidae), is a globally distributed species native to India. It was introduced for rearing in over 20 countries and regions including the Philippines, Egypt, Japan, and Australia during the 20th century. It was also introduced to China during the 1950s, where up to seven generations can occur each year [20,21]. It is a non-dormant, multivoltine insect and a suitable artificial host egg for *Trichogramma* parasitism. This is due to its advantages in terms of its low cost, high reproduction rate, short developmental period, and high resistance [22,23]. Some studies have found that ES eggs are used to rear *Trichogramma dendrolimi*, *Trichogramma confusum*, and *T. chilonis*. Therefore, maintaining an adequate population of female moths throughout the year is necessary to ensure a sufficient quantity of eggs for rearing egg parasitoids. Studies have shown that rearing *T. chilonis* on ES eggs has a benefit-to-cost ratio of 1.89, while the benefit-cost ratio of *T. chilonis* reared on *C. cephalonica* is 1.28 [24]. This indicates that ES eggs are a cost-effective option for mass rearing *T. chilonis*. Furthermore, research has demonstrated that using parasitized ES egg cards for one ha of land results in a 50% cost reduction compared to using *C. cephalonica* egg cards [25]. 

There is a limited amount of research regarding the parasitic impact of *Trichogramma* parasitoids on ES eggs, and no previous reports exist on *Trichogramma* wasps’ preference for parasitism regarding large eggs produced by the ES and COS. We examined the parasitism performance of these parasitoids toward these two hosts. *Trichogramma* have various parasitic adaptations to different factitious hosts; therefore, it is imperative to obtain suitable host eggs for efficient rearing in factories. Specifically, the variations in the nutrient uptake and developmental status of *Trichogramma* in the eggs of the two hosts are unclear. However, limited research has been conducted on *T. chilonis*’ preference for ES eggs and their offspring’s performance. In this context, we aimed to lay the foundation for the future exploration of the mass production of multiple species of *Trichogramma* using the eggs of the ES and COS through the biological study of the adaptive ability of *Trichogramma* reared on ES eggs and COS eggs. To evaluate the efficacy of egg parasitoids with respect to ES eggs, we assessed the parasitism performance of *T. chilonis* and their offspring with regard to the eggs of two factitious hosts.

## 2. Materials and Methods 

### 2.1. Hosts

The cocoons of COS were collected in Yongji City, Jilin Province, China, and stored in an incubator (4 °C, 70%, and natural photoperiod) for 2–3 months. They were then transferred to controlled indoor conditions (25 ± 1 °C, 60 ± 10% RH, and L12:D12 photoperiod) and hung to allow for the emergence of adult moths. New COS eggs used in the experiments were obtained by squeezing the abdomens of mature female moths that had been stored at 4 °C for 3–5 days, washed with distilled water, and air-dried. Immature green eggs were removed, and then large eggs were selected for experimental use under the laboratory conditions mentioned above [26].

The Sericultural Research Institute of the Chinese Academy of Agricultural Sciences provided ES cocoons, which were then reared in Guiyang City, Guizhou Province, China. The larvae were separately reared in a laboratory at 25 ± 2 °C with a relative humidity of 65 ± 5% and L14:D10 photoperiod. Newly hatched larvae were selected and transferred to rearing trays with a small brush and fed fresh castor leaves (*Ricinus communis* L.) until reaching the 5th instar stage. When rearing larvae on castor leaves, the 1st- and 2nd-instar larvae were fed tender leaves cut into small pieces. Medium-aged leaves were provided for 3rd-instar larvae, while mature leaves were given to 4th- and 5th-instar larvae. The silkworms were left to spit out silk, pupate, and form cocoons. Then, the cocoons were collected and hung on strings in a moth room (25 ± 1 °C, 60 ± 10% RH and natural photoperiod). After 10–11 days, unmated female moths were collected and stored at a temperature of 4 °C for approximately one week; this step was based on previous pre-test results showing that a maximal number of eggs matured during this period. For parasitoid rearing, newly emerged virgin female moths were used, and the eggs were extracted by gently compressing the moths’ abdomens. The eggs were washed with distilled water and air-dried [12].

### 2.2. Parasitoids

*Trichogramma chilonis* specimens were collected from Asian-corn-borer-parasitized eggs from maize fields in Changchun City, Jilin Province, China, in 2018. For the identification of *T. chilonis*, the morphology of male genital capsules and sequence analysis of rDNA-ITS2 (GeneBank Accession Nos. of HE648325 for *T. chilonis*) were utilized [27]. *Trichogramma* species were cultured for 10 generations on *C. cephalonica* eggs in laboratory conditions. The laboratory conditions were as follows: 25 ± 1 °C, 70 ± 5% relative humidity, and L14:D10 photoperiod.

### 2.3. Comparison of Basic Parameters of Two Host Eggs

To compare the impact of egg size and shell thickness on *T. chilonis*, we measured the egg length, width, and shell thickness of ES eggs and COS eggs using a microscope employing LAS v4.8 software. For eggshell thickness measurements, we embedded unfertilized and washed eggs into pre-made paraffin-containing molds at 56–58 °C and then sectioned them. (i) Wax block trimming: Cut and preserve the tissue around the excess paraffin to approximately 1–2 mm, ensuring the sides are parallel. (ii) Fixation of the paraffin block and angle adjustment: Secure the trimming wax block onto the slicer, and adjust the slicing blade and angle to 5°. (iii) Slicing: Adjust the slicing thickness to 3–5 μm. (iv) Slide preparation: Measure eggshell thickness using a digital microscope (Keyence, VHX-7000, Osaka, Japan) [28,29]. Each treatment consisted of 30 replicates.

### 2.4. Preferences of T. chilonis for Two Factitious Hosts

Parasitism experiments were performed on COS and ES eggs using *T. chilonis* specimens, which were reared on the eggs of *C. cephalonica*. The aim was to determine the preferences for the two factitious hosts, respectively. To ensure this was achieved, a non-toxic adhesive was used to attach five COS and ES eggs to a strip-shaped paper card, with a 1 cm spacing between the eggs. The eggs were transferred to a 3.5 cm diameter, 10 cm long glass tube along with a paper card. Five newly emerged *T. chilonis* adult females that had mated within 8 h were used in all experiments. After 24 h, the parental *T. chilonis* were removed, and the parasitized eggs were kept in an incubator at 25 ± 1 °C and 75 ± 5% RH and under an L14:D10 photoperiod to allow the parasitoids to develop. The eggs were monitored regularly under specified conditions for 5–6 days to record parasitoid development. After this period, parasitized eggs with a dark coloration were extracted individually from the paper card and transferred to separate glass tubes within the incubator. Each parasitized egg was observed daily until no more parasitoids emerged. *Trichogramma* emergence from the parasitized eggs occurs within 9 to 14 days, with a maximum of 15 days. The dates of parasitism and emergence, the number of dead wasps inside the eggs, female rate, emergence holes on each egg, pre-emergence time, and the number of emerged adults were recorded. Each treatment had 30 replicates.

### 2.5. Assessment of Parasitoid Body Size

Within 8 h of emerging, both male and female *T. chilonis* specimens reared on COS and ES were placed into a glass tube (1 cm diameter, 5 cm long). The *T. chilonis* specimens were then euthanized with chloroform. To estimate body size, 30 fresh specimens from each parasitoid were measured using the digital microscope (Keyence, VHX-7000), with three endpoints considered for each specimen. The measurements to be taken were (1) body length (from the margin of the forehead to the tip of the ovipositor), (2) head width (from the outer margin of left compound eye to the outer margin of right compound eye), and (3) hind tibia length [26]. 

### 2.6. Data Analysis

To compare the parasitism performance of *T. chilonis* with respect to the two factitious host eggs, the sizes of the offspring, and the size and eggshell thickness of the two factitious host eggs, all data were analyzed using Student’s *t*-test. All data were subject to normality testing via Shapiro–Wilk test prior to conducting Student’s *t*-test (*p* > 0.05). All statistical analyses were performed using SPSS 20.0 (SPSS Inc., Chicago, IL, USA).

## 3. Result

### 3.1. Egg Size and Eggshell Thickness of Two Factitious Hosts

In the present study, we examined differences in egg dimensions and shell thickness between two hosts, the COS and ES (Table 1), revealing statistically significant results. The results indicate that the COS eggs had significantly greater length, width, and shell thickness than the ES eggs (Length of eggs: *t* = 48.47, *df* = 58, *p* < 0.001, Shapiro–Wilk = 0.173, and 0.191 for the COS eggs and ES eggs; width of eggs: *t* = 57.81, *df* = 58, *p* < 0.001, Shapiro–Wilk = 0.273, and 0.243 for the COS eggs and ES eggs; shell thickness of eggs: *t* = 7.73, *df* = 58, *p* < 0.001, Shapiro–Wilk = 0.131, and 0.102 for the COS eggs and ES eggs).

### 3.2. Effects of Different Factitious Hosts on Parasitism Preference and Offspring Performance of Trichogramma chilonis

There were significant differences in the parasitism rates and emergence rates of *T. chilonis* with respect to the COS eggs and ES eggs (Figure 1). The parasitism rate of *T. chilonis* for the ES eggs was significantly higher than that for the COS eggs (*t* = 3.00, *df* = 58, *p* < 0.01, Shapiro–Wilk = 0.412, and 0.622 for the COS eggs and ES eggs). Similarly, the emergence rate of *T. chilonis* from the ES eggs was significantly higher than that from the COS eggs (*t* = 3.66, *df* = 58, *p* < 0.001, Shapiro–Wilk = 0.096, and 0.286 for the COS eggs and ES eggs).

Significant differences were observed in the number of emerged adults per egg, female rate, number of emergence holes, and pre-emergence time of *T. chilonis* when reared on two different factitious hosts (Table 2). The number of *T. chilonis* specimens reared on COS eggs was significantly higher than that on ES eggs (*t* = 9.05, *df* = 58, *p* < 0.001, Shapiro–Wilk = 0.501, and 0.678). There were no significant differences observed in the percentage of female offspring of *T. chilonis* across all egg treatments (*t* = 1.48, *df* = 58, *p* = 0.145, Shapiro–Wilk = 0.199, and 0.617). Moreover, the percentage of female progeny was over 80% in all treatments. However, the number of dead wasps in the COS eggs was significantly higher than that in the ES eggs (*t* = 3.97, *df* = 58, *p* < 0.01, Shapiro–Wilk = 0.077, and 0.112). There were differences in the number of emergence holes made by the parasitoid wasps emerging from different host eggs, with the number of emergence holes in the ES eggs being significantly higher than that in the COS eggs (*t* = 3.72, *df* = 58, *p* < 0.001, Shapiro–Wilk = 0.254, and 0.183). Significant differences were found in the pre-emergence times of *T. chilonis* for different factitious host eggs. The pre-emergence time was significantly higher for COS eggs compared to ES eggs (*t* = 8.86, *df* = 58, *p* < 0.001, Shapiro–Wilk = 0.237, and 0.205).

### 3.3. Effects of Different Factitious Hosts on the Body Size of Parasitoid Offspring

Rearing host species significantly affected the size of emerged *T. chilonis* wasps (Table 3). The body length of *T. chilonis* females emerged from COS eggs was significantly larger than that of *T. chilonis* females emerged from ES eggs (*t* = 3.37, *df* = 58, *p* < 0.01, Shapiro–Wilk = 0.184, and 0.927). The head width of *T. chilonis* females emerged from COS eggs was significantly larger than that of *T. chilonis* females emerged from ES eggs. (*t* = 3.47, *df* = 58, *p* < 0.001, Shapiro–Wilk = 0.630, and 0.538), The hind tibia length of *T. chilonis* females emerged from COS eggs was significantly larger than that of *T. chilonis* females emerged from ES eggs. (*t* = 2.78, *df* = 58, *p* < 0.01, Shapiro–Wilk = 0.268, and 0.795). Similarly, there were significant differences in body length, head width, and hind tibia length between male *T. chilonis* reared on COS eggs and male *T. chilonis* reared on ES eggs. The body length of *T. chilonis* males reared on COS eggs was significantly larger than that of *T. chilonis* males reared on ES eggs (*t* = 12.54, *df* = 58, *p* < 0.001, Shapiro–Wilk = 0.927, and 0.194). The head width of *T. chilonis* males reared on COS eggs was significantly larger than that of *T. chilonis* males reared on ES eggs (*t* = 5.72, *df* = 58, *p* < 0.001, Shapiro–Wilk = 0.538, and 0.499). The hind tibia length of *T. chilonis* males reared on COS eggs was significantly larger than that of *T. chilonis* males reared on ES eggs (*t* = 5.07, *df* = 58, *p* < 0.001, Shapiro–Wilk = 0.881, and 0.739).

## 4. Discussion

The research and development of cost-effective methods for the mass production of *Trichogramma* are crucial to biological control [1]. We investigated the parasitism preferences of *T. chilonis* for two factitious hosts labeled “large eggs”. We compared the production efficiency of *T. chilonis* reared on the two factitious host eggs under the same conditions and recorded the following parameters: parasitism rate, emergence rate, female rate, pre-emergence time, emergence holes, and the number of adults that emerged per egg. We found that the parasitism and suitability of the *T. chilonis* specimens that emerged from ES eggs were better than those of the specimens that emerged from COS eggs. The *Trichogramma* specimens reared on ES eggs displayed a higher parasitism rate, emergence rate, shorter pre-emergence time, and fewer deaths among the wasps. Our study also revealed that there were significant differences in the body sizes of the female and male offspring of *T. chilonis* when reared on two factitious hosts. This experiment demonstrates that *T. chilonis* can be effectively raised using ES eggs as a host, leading to new directions for *Trichogramma* production factories, particularly with regard to the cultivation of factitious hosts for these parasitoids. Host species significantly impact parasitoid parasitism and emergence success, including with regard to physical attributes like shape, size, texture, and color, as well as concerning chemical characteristics like substances linked to host eggs [30]. Parasitism and emergence rates are critical metrics for assessing the suitability of potential hosts [31,32], notably for mass production [1]. The rates of parasitism and emergence for *T. chilonis* reared on ES eggs were notably higher than those reared on COS eggs, and this factor was influenced by host species. Some studies suggest that the eggshell structure of factitious host eggs, including their surface structure and eggshell thickness, can limit the parasitism of certain species of *Trichogramma*, such as *Trichogramma japonicum* and *Trichogramma ostriniae*. These species have short ovipositors and therefore cannot puncture the eggshells of COS eggs (a factitious host) [33,34]. We suggest that the hardness and thickness of COS eggshells may contribute to this phenomenon. As a result, some *Trichogramma* females’ ovipositors may not be able to penetrate eggshells. Even if they are successful in laying their offspring inside the eggs, the offspring may not be able to bite through the eggshells and emerge successfully [35,36]. In our study, we observed that the eggshell thickness of the ES eggs was approximately 15 μm thinner than that of the COS eggs. Thus, the difference in eggshell thickness between the two species may be a significant factor contributing to dissimilarities in the parasitism and emergence rates of *T. chilonis*.

There are no significant differences in the proportions of female offspring among various host species used for rearing *T. chilonis* via parasitism. COS eggs have been found to be effective for mass producing a single *Trichogramma* species [12], as they consistently produce a higher number of female wasps, with a female ratio of up to 90%. Therefore, the number of female wasps among the offspring is also significantly higher [37]. The reproductive capability of *Trichogramma* offspring reared using ES eggs is comparable to that of *Trichogramma* parasitoids reared using COS eggs.

This study concluded that significantly more *T. chilonis* parasitoids could be reared using a single COS egg than using a single ES egg and that, on average, each COS egg produced approximately 35 more *T. chilonis* individuals than each ES egg, which may be due to the effect of host size. To optimize their offspring’s fitness for parasitism, female parasitoid wasps adjust the number of offspring they produce according to the host encountered [38,39]. The size of the host can impact the fitness and survival of parasitized offspring [40,41,42]. Additionally, parasitoid wasp females have the ability to modify the number of eggs laid according to the host’s size [43]. COS eggs are the most readily available and efficient mass-production hosts for parasitoid wasps in laboratory conditions. The use of large eggs in mass production systems has demonstrated practical advantages. Additionally, employing ES and COS eggs is more cost-effective for rearing *Trichogramma* parasitoids compared to rice moth eggs [44,45,46].

Generally, larger host eggs allow for the rearing of larger egg parasitoids, as demonstrated by the results of this study. The body length, head width, and hind tibia length of *T. chilonis* reared on COS eggs were greater than those of *T. chilonis* reared on ES eggs. Furthermore, larger female parasitoid wasps exhibited increased parasitism, oviposition, and host-searching abilities [47,48]. According to previous reports, egg parasitoids that are reared on larger host eggs are indeed larger. However, the unsuitability of the host resource may have a more significant impact on the development of wasps, resulting in females clutching fewer eggs [49,50]. Therefore, the screening of suitable hosts can enhance the quality of egg parasitoid products [51].

Our study reveals lower mortality of *T. chilonis* offspring reared on ES eggs compared to COS eggs. This difference can be attributed to the varying resources available for each egg as a result of the diverse host populations and to the fact that higher numbers of wasps in an egg may result in parasitoid wasps competing for nutrients within the egg [52,53]. When host nutrient resources are limited, offspring wasps consume only a small portion to ensure complete growth. This results in smaller or deformed individuals and may even prevent some parasitoids from successfully emerging from their eggs, ultimately lowering survival rates [54,55,56]. Thus, the quantity of dead *T. chilonis* wasps was elevated in the COS eggs, while it was limited in the ES eggs. For parasitoid wasps, the recognition and acquisition of nutrients from their host are crucial factors for the survival of their offspring. To ensure the survival of its offspring, a parasitoid wasp must ensure that various physiological and nutritional adaptations take place [57]. Consequently, the size of the host does not necessarily influence its suitability as a host.

Additionally, the number of emergence holes is linked to the emergence of parasitic wasps. Our previous study revealed that certain *T. chilonis* wasps were unable to successfully create or pass through emergence holes due to their oversized bodies or lack of energy, resulting in blocked holes and an increase in wasp mortality rates while decreasing the overall emergence rate. Therefore, based on our study, indicating a higher number of emergence holes in ES eggs compared to COS eggs (which could be a result of thicker eggshells), increasing the number of paths for offspring wasps to emerge from led to a higher emergence rate and a lower death rate for the wasps.

Pre-emergence time is a crucial factor in the mass production of *Trichogramma*. Shorter developmental time in factory production can reduce labor and time costs, making rearing at a mass production scale more efficient [1,14]. In this study, it was found that the development times of *T. chilonis* in ES eggs were significantly faster than those in COS eggs.

In summary, the ES is globally distributed and can be reared indoors, similar to the silkworm *Bombyx mori*. This study confirms that the ES is suitable for use in the mass production of *T. chilonis*, and its eggs provide more significant benefits than those of the COS when used as factitious hosts for the mass rearing of *Trichogramma* wasps worldwide. *Trichogramma* wasps produced via the parasitism of the eggs of the ES have greater potential in terms of promoting the application of the control of pests in agriculture and forestry. 

## Figures and Tables

**Figure 1 insects-15-00002-f001:**
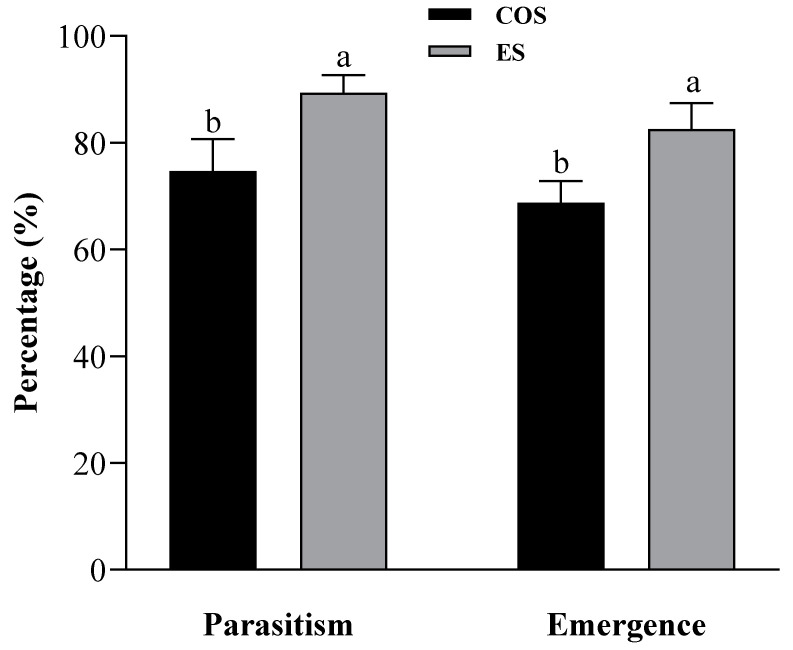
Percentages of parasitism and emergence of (means ± SEs) *T. chilonis* for eggs of COS and ES. Different lowercase letters above the bars indicate significant differences between parasitoid wasps for different factitious host eggs (Student’s *t*-test, *p* < 0.05).

**Table 1 insects-15-00002-t001:** Egg size and shell thickness of the Chinese oak silkworm (COS) and the eri silkworm (ES).

Parameters	Host	
	COS	ES
Egg length (μm)	2908.80 ± 22.66 a	1721.48 ± 9.31 b
Egg width (μm)	2574.39 ± 8.41 a	1178.85 ± 6.42 b
Eggshell thickness (μm)	47.30 ± 1.64 a	33.77 ± 0.62 b

For each parameter, means ± SEs are shown. Each parameter value followed by different lower-case letters in the rows was significantly different based on the results obtained using Student’s *t*-test.

**Table 2 insects-15-00002-t002:** Biological parameters of *T. chilonis* with respect to eggs of the Chinese oak silkworm (COS-Tc) and the eri silkworm (ES-Tc).

Parameters	Parasitoids Reared on Different Factitious Hosts	
	ES-Tc	COS-Tc
Number of adults that emerged	29.03 ± 1.79 b	65.37 ± 3.60 a
Dead wasps in host egg	0.77 ± 0.25 b	7.27 ± 1.62 a
Female rate (%)	86.53 ± 1.23 a	83.68 ± 1.49 a
Emergence holes	1.63 ± 0.09 a	1.20 ± 0.07 b
Pre-emergence time (day)	10.89 ± 0.13 b	12.59 ± 0.14 a

For each parameter, means ± SEs are shown. Each parameter value followed by different lower-case letters in a row is significantly different between two different hosts (Student’s *t*-test, *p* < 0.05).

**Table 3 insects-15-00002-t003:** Comparison of body size of females and males of the offspring of *T. chilonis* reared on eggs of COS (COS-Tc) and ES (ES-Tc).

Sex	Parameters	COS-Tc	ES-Tc
Female	Body length	626.92 ± 6.50 a	578.83 ± 12.69 b
	Head width	282.33 ± 8.82 a	247.50 ± 4.80 b
	Hind tibia length	219.79 ± 3.51 a	194.92 ± 8.24 b
Male	Body length	556.69 ± 7.28 a	438.43 ± 5.99 b
	Head width	242.57 ± 4.73 a	210.33 ± 3.07 b
	Hind tibia length	178.37 ± 3.46 a	155.46 ± 2.91 b

For each parameter, means ± SEs are shown. Each parameter value followed by different lower-case letters in a row is significantly different according to the Student’s *t*-test (*p* = 0.05).

## Data Availability

The data presented in this study are available on reasonable request from the corresponding author.

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
