# Peer review of "Parasitism and Suitability of Trichogramma chilonis on Large Eggs of Two Factitious Hosts: Samia cynthia ricini and Antheraea pernyi"

_insects, 2023, doi:10.3390/insects15010002_

Round 1

Reviewer 1 Report

Comments and Suggestions for Authors

This study evaluates Trichogramma chilonis for lepidopteran pest control, emphasizing mass production using ES eggs. The results highlight its effectiveness on both COS and ES eggs, crucial for cost-effective Trichogramma rearing. While endorsing publication, a significant revision is suggested to address manuscript constraints.

A notable limitation is found in the analysis, particularly the use of ANOVA or unspecified approaches for percentage parasitism data, considered inappropriate. Proportion data, ranging from 0% to 100%, calls for alternative methods like logistic GLM or a binomial distribution (refer to Ecology 2011 92, 3-10). These methods better handle the bounded nature, preventing predictions beyond limits. Logistic regression, widely accepted, converges appropriately at 0 and 1, avoiding the risk of predicting negative or over-100% values. I recommend revisiting the Results and Discussion sections to align them with these considerations.

The critique extends to data transformation procedures, emphasizing the importance of examining residuals using untransformed data initially. Consider arcsine transformation only if residuals do not meet assumptions. The merits of arcsine transformation are questioned in the literature, favoring logistic regression for binomially distributed proportional data, ensuring biological relevance, data within 0 and 1, and interpretable coefficients. The potential for biased results with arcsine transformation warrants reconsideration of the data analysis approach in the results and discussion sections.

It is insufficient to state that all data were analyzed using ANOVA or GLMs; clarity is needed on dependent and independent variables, any transformations, or their absence. The manuscript lacks clarity on parameters investigated and why transformations were applied or if the data were normally distributed.

The Results and Discussion sections require improvement in clarity and conciseness, focusing on results from this specific study. A revision is recommended to address these concerns before resubmitting the manuscript.

Author Response

This study evaluates Trichogramma chilonis for lepidopteran pest control, emphasizing mass production using ES eggs. The results highlight its effectiveness on both COS and ES eggs, crucial for cost-effective Trichogramma rearing. While endorsing publication, a significant revision is suggested to address manuscript constraints.

A notable limitation is found in the analysis, particularly the use of ANOVA or unspecified approaches for percentage parasitism data, considered inappropriate. Proportion data, ranging from 0% to 100%, calls for alternative methods like logistic GLM or a binomial distribution (refer to Ecology 2011 92, 3-10). These methods better handle the bounded nature, preventing predictions beyond limits. Logistic regression, widely accepted, converges appropriately at 0 and 1, avoiding the risk of predicting negative or over-100% values. I recommend revisiting the Results and Discussion sections to align them with these considerations.

The critique extends to data transformation procedures, emphasizing the importance of examining residuals using untransformed data initially. Consider arcsine transformation only if residuals do not meet assumptions. The merits of arcsine transformation are questioned in the literature, favoring logistic regression for binomially distributed proportional data, ensuring biological relevance, data within 0 and 1, and interpretable coefficients. The potential for biased results with arcsine transformation warrants reconsideration of the data analysis approach in the results and discussion sections.

It is insufficient to state that all data were analyzed using ANOVA or GLMs; clarity is needed on dependent and independent variables, any transformations, or their absence. The manuscript lacks clarity on parameters investigated and why transformations were applied or if the data were normally distributed.

The Results and Discussion sections require improvement in clarity and conciseness, focusing on results from this specific study. A revision is recommended to address these concerns before resubmitting the manuscript.

 Answer: Dear Editor, we used Student’s t-test to analyzed and compared the parameters of the parasitism performance of Trichogramma chilonis on the two factitious host eggs. We also used the Student’s t-test for the size of the offspring, and the size and eggshell thickness of the two factitious host eggs. We double-checked the data before the analysis and found that all the data meet assumptions, so we omitted the arcsine this time. At the same time, we re-modified the corresponding position in the article. Thank you very much for your advice. See line 199-203, page 5.

Reviewer 2 Report

Comments and Suggestions for Authors

I have read with attention the ms titled: "Parasitism and suitability of Trichogramma chilonis on large eggs of two factitious hosts: Samia cynthia ricini and Antheraea pernyi ", and authored by Yue-Hua Zhang, Ji-Zhi Xue, Talha Tariq, Tian-Hao Li, He-Ying Qian, Wen-Hui Cui, Hao Tian, Lucie S. Mon- ticelli, Nicolas Desneux3 and Lian-Sheng Zang.

Below you will find my remarks, comments, and suggestions. 

 Line 38-40: Please consider replacing ‘body size’ with ‘body length’

Line 46 Please replace technology with method

Line 48 I suggest using plurals as the verb

Line 72. Abbreviate the Ephestia

Line 80. Replace polyphagous with generalist

Line 97: The author provided authorities information for a few species not for all. Please be consistent. Also provide the taxonomic classification including order and family for each species at their first mention in the main text. Refer to the journal's guidelines for more detailed instructions on taxonomic notation.

101-102: Please clarify for which species of Trichogramma, the Eri silkworm is considered a suitable host?

Line 113: it might be clearer to specify 'large eggs of ES and COS' instead of 'two larger eggs.

Line 116-117: Author mentioned “the variations in nutrient uptake and developmental status of Trichogramma in the eggs of two hosts”, yet it appears this aspect was not included in your study. Could you please clarify the context or relevance of this statement within the scope of your research?"

Line 126-127: Please provide the relative humidity and photoperiodic condition of incubator that was set at 4 °C.

Line 127-129: Please rephrase the sentence. Also provide photoperiodic condition

Line 129: We never start sentence with abbreviation

Line 131-132: Please define the criteria used to determine the 'health' of eggs.

Line 136-137: Please specify the photoperiodic condition used

Line 155: Please remove the word 'temperature' as '°C' already implies temperature

Line 166-167: please specify if the same Keyence VHX-2000 digital microscope was used for measuring both egg length and width as well as eggshell thickness.

Line 168-181: Please specify which parameters were collected in this experiment

Line 180-181: Please provide the range and the maximum number of days for wasp emergence from the parasitized eggs.

Line 213-217: In figure 1, Author referred to the host as 'COS' and 'ES', while in the figure bar, their scientific names were used. For consistency and clarity, it would be better to use either the common names or the scientific names consistently in both the legend and the figure. Also check the journal guidelines

 Line 208-231: In the results section, the outcomes of the second experiment are divided under two separate headings, which creates some confusion. Considering that several parameters were measured in this experiment, it would enhance clarity if the results were consolidated under a single, comprehensive heading or organized in a way that clearly delineates the different aspects of the experiment.

 Line 239. Please remove the period between 'ES eggs' and the bracket.

 Line 236-250: I've observed that the term 'reared' is frequently used in the section discussing the size variations of T. chilonis when reared on COS and ES eggs. To enhance the text's variety and readability, the author might consider using alternative terms where appropriate. For example, 'The body length of T. chilonis males emerged from COS eggs was significantly larger than that of T. chilonis males emerged from ES eggs.' This change could provide a clearer distinction between the process of rearing and the outcome (emergence) of that process.

Line 197-254: The data in the manuscript is redundantly presented in figures, tables, and text. It's recommended to avoid this repetition: use figures and tables for visual data presentation, and text for interpretation and context, not for reiterating detailed data.

Line 261-264: Please consider revising the text for clarity. 

Line 370-371: Please check the formatting of reference number 5. Initial letters of each major word in the paper title are capitalized. However, this capitalization style is not consistent with other references.

Author Response

I have read with attention the ms titled: "Parasitism and suitability of Trichogramma chilonis on large eggs of two factitious hosts: Samia cynthia ricini and Antheraea pernyi ", and authored by Yue-Hua Zhang, Ji-Zhi Xue, Talha Tariq, Tian-Hao Li, He-Ying Qian, Wen-Hui Cui, Hao Tian, Lucie S. Mon- ticelli, Nicolas Desneux3 and Lian-Sheng Zang.

Below you will find my remarks, comments, and suggestions. 

 Line 38-40: Please consider replacing ‘body size’ with ‘body length’

Answer: DONE. See line 38-40, page 1.

Line 46 Please replace technology with method

Answer: DONE. See line 46, page 2.

Line 48 I suggest using plurals as the verb

Answer: Thank you for your suggestion. We have rewritten this sentence (line 47-51)

Line 72. Abbreviate the Ephestia

Answer: DONE. See line 76, page 2.

Line 80. Replace polyphagous with generalist

Answer: DONE. See line 84, page 2.

Line 97: The author provided authorities information for a few species not for all. Please be consistent. Also provide the taxonomic classification including order and family for each species at their first mention in the main text. Refer to the journal's guidelines for more detailed instructions on taxonomic notation.

Answer: We thank the reviewer for the positive comments and suggestions. We have supplemented the species information. See line 101, page 3.

101-102: Please clarify for which species of Trichogramma, the Eri silkworm is considered a suitable host?

Answer: DONE. See line 107-109, page 3.

Line 113: it might be clearer to specify 'large eggs of ES and COS' instead of 'two larger eggs.

Answer: Good suggestion. We have changed “two larger eggs” to “large eggs of ES and COS. See line 118, page3.

Line 116-117: Author mentioned “the variations in nutrient uptake and developmental status of Trichogramma in the eggs of two hosts”, yet it appears this aspect was not included in your study. Could you please clarify the context or relevance of this statement within the scope of your research?"

Answer: Thank you for your query. Some studies suggest significant differences in the suitability of different hosts for Trichogramma, which may be due to differences in the nutrients in the eggs. Additionally, the nutrients in the eggs affect Trichogramma species' growth and lead to changes in their body size.

Line 126-127: Please provide the relative humidity and photoperiodic condition of incubator that was set at 4 °C.

Answer: Thanks for your comments. We have provided the conditions for the incubator. See line 132, page 3.

Line 127-129: Please rephrase the sentence. Also provide photoperiodic condition

Answer: Thank you for your suggestion. We have rephrased the sentence and provided the incubator conditions. See line 133-134, page 3.

Line 129: We never start sentence with abbreviation

Answer: DONE. See line 135, page 3.

Line 131-132: Please define the criteria used to determine the 'health' of eggs.

Answer: The COS eggs we used earlier were obtained by squeezing the abdomen of mature female moths, washed with distilled water, and air-dried. The eggs were tested after removing immature green eggs. We apologize for the ambiguity caused by the expression in the manuscript, and we have made changes in the manuscript. See line 137, page 3.

Line 136-137: Please specify the photoperiodic condition used

Answer: Thank you for your suggestion. We have provided the photoperiodic conditions. See line 142, page 3.

Line 155: Please remove the word 'temperature' as '°C' already implies temperature

Answer: DONE. See line 160, page 4.

Line 166-167: please specify if the same Keyence VHX-2000 digital microscope was used for measuring both egg length and width as well as eggshell thickness.

Answer: We determined the egg length, egg width and eggshell thickness measured using the same Keyence VHX-2000 digital microscope.

Line 168-181: Please specify which parameters were collected in this experiment

Answer: DONE. See line 187-189, page 4.

Line 180-181: Please provide the range and the maximum number of days for wasp emergence from the parasitized eggs.

Answer: DONE. See line 186-187, page 4.

Line 213-217: In figure 1, Author referred to the host as 'COS' and 'ES', while in the figure bar, their scientific names were used. For consistency and clarity, it would be better to use either the common names or the scientific names consistently in both the legend and the figure. Also check the journal guidelines

Answer: Thank you for your suggestion. We have used common names in figure 1 that are consistent with the manuscript. See line 227-231 page 5-6.

 Line 208-231: In the results section, the outcomes of the second experiment are divided under two separate headings, which creates some confusion. Considering that several parameters were measured in this experiment, it would enhance clarity if the results were consolidated under a single, comprehensive heading or organized in a way that clearly delineates the different aspects of the experiment.

Answer: DONE. See line 220-221, page 5.

 Line 239. Please remove the period between 'ES eggs' and the bracket.

Answer: DONE. See line 253, page 6.

 Line 236-250: I've observed that the term 'reared' is frequently used in the section discussing the size variations of T. chilonis when reared on COS and ES eggs. To enhance the text's variety and readability, the author might consider using alternative terms where appropriate. For example, 'The body length of T. chilonis males emerged from COS eggs was significantly larger than that of T. chilonis males emerged from ES eggs.' This change could provide a clearer distinction between the process of rearing and the outcome (emergence) of that process.

 Answer: DONE. See line 251-257, page 6-7.

Line 197-254: The data in the manuscript is redundantly presented in figures, tables, and text. It's recommended to avoid this repetition: use figures and tables for visual data presentation, and text for interpretation and context, not for reiterating detailed data.

Answer: Thanks for your comments. The figures and tables represent distinct parameters, while the text provides crucial data. We believe that the arrangement of the figures and tables helps the reader to see the differences between them.

Line 261-264: Please consider revising the text for clarity. 

 Answer: We apologize for not providing clear information on this section. I have revised this sentence. See line 276-280, page 7.

Line 370-371: Please check the formatting of reference number 5. Initial letters of each major word in the paper title are capitalized. However, this capitalization style is not consistent with other references.

Answer: Thank you for your revision suggestions. We have rechecked the format of reference number 5 and made changes. See line 388-389, page 9.